# Hand Gesture Recognition Using Compact CNN via Surface Electromyography Signals

**DOI:** 10.3390/s20030672

**Published:** 2020-01-26

**Authors:** Lin Chen, Jianting Fu, Yuheng Wu, Haochen Li, Bin Zheng

**Affiliations:** 1Chongqing Institute of Green and Intelligent Technology, Chinese Academy of Sciences, Chongqing 400700, China; chenlin18@cigit.ac.cn (L.C.); jiantingfu@cigit.ac.cn (J.F.); wuyuheng21@163.com (Y.W.); lihaochen@csu.ac.cn (H.L.); 2School of Computer Science and Technology, University of Chinese Academy of Sciences, Beijing 100049, China; 3School of Mechatronical Engineering, Changchun University of Science and Technology, Changchun 130022, China

**Keywords:** surface electromyography (sEMG), convolution neural networks (CNNs), hand gesture recognition

## Abstract

By training the deep neural network model, the hidden features in Surface Electromyography(sEMG) signals can be extracted. The motion intention of the human can be predicted by analysis of sEMG. However, the models recently proposed by researchers often have a large number of parameters. Therefore, we designed a compact Convolution Neural Network (CNN) model, which not only improves the classification accuracy but also reduces the number of parameters in the model. Our proposed model was validated on the Ninapro DB5 Dataset and the Myo Dataset. The classification accuracy of gesture recognition achieved good results.

## 1. Introduction

In recent years, Surface Electromyography(sEMG) signals have been widely used in artificial limb control, medical devices, human-computer interaction, and other fields. With the development of artificial intelligence and robotics technology, the intention of human hand movements can be obtained by using an artificial intelligence algorithm to analyze the sEMG signals collected from the residual limb. Robotics and artificial intelligence can be leveraged to better help the disabled people to independently complete some basic interactions in their daily life. The sEMG signals, which are non-stationary, represent the sum of subcutaneous athletic action potentials generated through muscular contraction [1]. Also, it is one of the main physical signals of an intelligent algorithm to identify motion intention.

Distinguishing sEMG signals collected from different gestures is the core part of the related applications using sEMG signals as intermediate media. At present, the literature on gesture recognition or artificial limb control by sEMG signals primarily focuses on the time and frequency domain feature extraction of sEMG signals, which aims to distinguish sEMG signals by feature recognition [1,2,3]. After years of exploration by researchers, some effective feature combinations have been proposed in both the time domain and frequency domain [4,5,6], and some fruitful results have been achieved with their respective datasets. Choosing feature extraction is particularly important in that different gestures can be distinguished by traditional methods. However, it is difficult to improve the performance of gesture recognition based on sEMG by traditional methods. Nevertheless, the process of designing and selecting features can be complicated and the combinations of features are diverse, leading to increasing of workload and dissatisfied results [7].

Using deep neural networks to distinguish sEMG signals has been proposed by researchers. Wu et al. [7] proposed LCNN and CNN_LSTM models (These models can be thought of as autoencoders for automatic feature extraction.), which do not require the process of traditional feature extraction. In recent years, deep learning has achieved great success in the field of image recognition. An important idea was put forward in [8,9] that the signals of a channel can form a graph, after the short- time Fourier transform or wavelet transform of sEMG signals. It was a good idea to convert the sEMG signal into an image and inspired us with the transform of the sEMG signal. Researchers such as Côté-Allard et al. [8], who regarded the original sEMG signals as an image, constructed the ConvNet model to further improve the classification accuracy of sEMG signals. However, the LCNN and CNN_LSTM models proposed by Wu et al. [7], and the ConvNet model used by Côté-Allard et al. [8], contain a large number of parameters.

For deep learning algorithms, the final test accuracy is directly affected by the size of the training data, but one participant cannot be expected to generate tens of thousands of examples in one experiment during the data collection process. However, a large amount of data can be obtained by aggregating the records of multiple participants, so that the model can be pre-trained to reduce the amount of data required by new participants. On the other hand, designing a compact network structure to reduce the number of parameters can also reduce the demand for data size.

In order to reduce the number of model parameters and improve the accuracy of model classification, we present a new compact deep convolutional neural network model for gesture recognition, called as EMGNet. It was validated on the Myo Dataset that the average recognition accuracy of EMGNet can achieve 98.81%. The NinaPro DB5 dataset has often been used to test classical machine learning methods. The accuracy of the EMGNet on these datasets was higher than that of the traditional machine learning methods. Figure 1 shows the overall flow chart of sEMG signal acquisition and identification. 

The rest of this paper is organized as follows. The related work of gesture recognition through deep learning is outlined in Section 2. The data processing and network architecture are described in detail in Section 3. The proposed network model is compared with the current excellent deep network framework and the classical machine learning classification method in Section 4. Finally, we present the conclusion in Section 5.

## 2. Related Work

People will produce different signals when completing the same action even lots of precise control electrodes are used to sense them [10]. Therefore, it is difficult to recognize sEMG signals. Since the AlexNet network proposed by Krizhevksy et al. [11] won the ImageNet challenge in 2012, deep learning has achieved great success in image classification, speech recognition, and other fields. Images can be accurately classified by training the neural network model to learn the characteristics of images. Nowadays, exploring network architecture has become part of deep learning research. 

Currently, some researchers have successfully applied deep learning to sEMG signal classification and explored several effective network frameworks [8,12,13,14,15,16]. Using CNN to classify sEMG signals, literature [12,17,18] took the raw signals as input space. The spectrograms of raw sEMG signals were obtained by Short-Time Fourier Transform (STFT) and fed into the convolutional network (ConvNets) [13,19]. Literature [8] used the ConvNets to classify the characterizations of the sEMG signals extracted by short-time Fourier transform-based spectrogram and Continuous Wavelet Transform (CWT). Since sEMG signals correspond to the timing signal, we proposed to classify the sEMG signal by combining Long Short-Term Memory (LSTM) and CNN from our previous work [7]. The temporal information in the signal is retained and the ability of CNN to extract features is utilized. We take advantage of the complementarity of CNNs and LSTMs by combing them into one unified architecture. Meanwhile, we analyze the effect of adding CNN before the LSTM. We propose LCNN and CNN-LSTM models, which can directly input pre- processed EMG signals into the network [7]. In practical work, we verified that the performance of the LCNN model is better than CNN-LSTM. Figure 2 simply depicts the architecture of the LCNN model. We use PReLU [20] as the non-linear activation function. ADAM [21] is utilized for the optimization of the model.

However, the ConvNets model (Shown in Figure 3) in [8] was too complicated, and the LSTM model was introduced in [7], which led to expensive computation in gesture recognition. Therefore, a new network model was proposed in this paper, and it was proved by experiment that this model not only improves the accuracy of recognition, but also reduces the complexity of the network model.

## 3. sEMG Signals Recognition Algorithm

### 3.1. sEMG Signals Feature Extraction

Designing a sEMG signal feature is one of the main tasks of the algorithm. It can not only easily distinguish the sEMG signals produced by different movements, but also maintain a small variance between the sEMG signals produced by the same movements. Four feature sets—Time domain characteristics (TD) [5], Enhanced TD [4], Nina Pro Features [6,22], and SampEn Pipeline [23]—were selected as features of the classical machine learning algorithm compared to the proposed method. The TD feature set includes four features: Mean Absolute Value (MAV), Zero Crossing (ZC), Slope Sign Changes (SSC), and Waveform Length (WL) [5]. The Enhanced TD includes features such as Root Mean Square (RMS) and Autoregressive Coefficient (AC) [3]. Nina Pro Features includes Root Mean Square, TD features, etc. [6,22]. SampEn Pipeline includes features such as Sample Entropy (SampEn) [24], Root Mean Square, and Waveform Length [23]. 

Since the sEMG signals is a non-stationary signal [25], the analysis of these signals using the Fourier transform is limited. One technique to solve this problem is the Short Time Fourier Transform (STFT), which separates the signal into smaller segments by applying a sliding window and separately calculates the Fourier transform for each of the segments. The spectrogram of the signal can be calculated from the square of the signal after STFT transformation. When the signal *x*(*t*) and the window function *w*(*t*) are given, the spectrogram is calculated as follows:(1)$$spectrogram\;\left(x(t),w(t)\right)={\left|STFT_{x}(t,f)\right|}^{2}$$
(2)$$STFT_{x}\left(t,f\right)=\int_{-\infty }^{+\infty }\left[x(u)w(u-t)\right]e^{-j2\pi fu}du$$
where f represents frequency. The wavelet transform(WT) is similar to the STFT, but it overcomes the shortcomings of the window in the STFT which does not change with frequency. The WT adapts to the change in frequency in the signal by adjusting the width of the window. When the frequency in the signal climbs high, the WT increases the resolution by narrowing the time window. In addition, WT, which is an ideal signal analysis tool, has the ability to obtain the amplitude and frequency of mutations in a signal.
(3)$$X(a,b)=\frac{1}{\sqrt{b}}\int_{-\infty }^{\infty }x\left(t\right)\varphi \left(\frac{t-a}{b}\right)dt$$
(4)$$\int_{-\infty }^{+\infty }\frac{{\left|\varphi (\omega )\right|}^{2}}{\omega }d\omega <\infty$$
where the Fourier Transform *φ*(*ω*) of *φ*(*t*) must satisfy Equation (4). *φ*(*t*), also known as the mother wavelet function, is a signal with a limited duration, varying frequency and a mean of zero. The scaling factor *b* controls the scaling of the wavelet function. The translation amount *a* controls the translation of the wavelet function.

After the continuous wavelet transform of the sEMG signal, the corresponding spectrum information will be obtained, which is similar to the image in scale and also contains the frequency-domain information of the timing sequence data. The datasets tested in this article were all collected by Myo armband, as shown in Figure 4. Myo is an 8-channel, dry-electrodes, low-sampling rate (200 Hz), low-cost consumer grade sEMG armband, which is convenient to wear and easy to use [7]. The data of one channel was separated by applying sliding windows of 52 samples (260 ms). The mother wavelet of continuous wavelet transform adopts the Mexican Hat wavelet function. The CWTs were calculated with 32 scales obtaining a 32 × 52 matrix. The scale (32 × 52) sampled down 0.5 in spectrum information is taken as the input of EMGNet model. Thus the input of the EMGNet model has 8 channels, each of which consists of a matrix of size 15 × 25. Figure 5b is the spectrum of the signal shown in Figure 5a after wavelet transformation.

### 3.2. EMGNet Architecture

A new network model called EMGNet is proposed in this paper, and shown in Figure 6. It consists of four convolutional layers and a max pooling layer without using the full connection layer as the final output. ConvNet architecture (Shown in Figure 3) contains 67179 learnable parameters used in the CWT+TL method proposed in Literature [8]. As shown in Table 1, the model proposed in this paper contains fewer parameters than the models used in current advanced methods. In view of the better performance of the EMGNet model in the actual measurement, this section mainly introduces the EMGNet model.

The cost function Loss can be computed as follows:(5)$$Loss=-\sum_{i=1}^{n}y_{i}\log\left(y_{i}{}^{\prime }\right)$$
where *y_i_* is the truth value of the *i*th category, *n* is the number of categories, and *y_i_’* is the ith category predicted value of the output. Because we adopted One-Hot Encoding, the true value of the one category is 1 while the others are 0. 

The advanced optimization method is used for the backpropagation of the EMGNet, and our final target is to minimize the cost function Loss. In the field of image recognition, the size of convolution kernels commonly used by researchers is 3 × 3 and 5 × 5 [11,26,27]. We found that the experimental results obtained by setting the size of the convolution kernel to 3 × 3 are better. Therefore, the size of all convolution kernels in the EMGNet model is set to 3 × 3. Meanwhile, in order to reduce the parameters of the model as much as possible, the feature map of each layer of the EMGNet model is also set smaller. When the feature map of the network model increases, the step length in the convolution process is set to 2, so as to achieve the effect of halving. In other cases, the step length and the padding are set to 1, so that the feature remains unchanged. In order to further reduce the number of network parameters, the output of the model does not use the full connection layer, but carries out adaptive mean sampling first, and then uses the convolution layer for classification. 

## 4. Experiment and Results

We evaluated our mode on two publicly available hand gesture recognition datasets composed of Myo Dataset [8,9] and NinaPro DB5 Dataset [6]. First, we compared the method proposed with the methods of classical machine learning and three other methods (CNN-LSTM, LCNN, CWT+TL) on the Myo Dataset [8,9]. Then, it was compared with the methods of classical machine learning and the methods (CNN-LSTM, LCNN) proposed earlier on the NinaPro DB5 Dataset [6].

### 4.1. Evaluation Dataset

Containing two different sub-datasets, this gesture dataset (Myo Dataset [8,9]) was collected using the Myo armband. The first Dataset was used as the pre-training dataset in [8] and the second includes the part of training and testing. The former is mainly used to establish, verify, and optimize the classification model, which consists of 19 subjects. The latter, used only for final training and validation, consists of 17 participants. The second Dataset contains a training section and two test sections in the literature [8,9], which is an unreasonable arrangement and leads to a decrease in the amount of training data. In order to facilitate the comparative experiment, this article uses the same settings. The Myo Dataset contains 7 types of gestures, and there are significant differences between gestures. It provides sufficient amount of data, and its gestures are shown in Figure 7.

This dataset (NinaPro DB5 [6]) is based on benchmark sEMG-based gesture recognition algorithms [6] containing data from 10 able-bodied participants divided into three exercise sets—Exercise A, B, and C contain 12, 17, and 23 different movements (including neutral) respectively. It is recorded by two Myo armbands, and only one of them is used in this work. One of the characteristics is that there are some similar gestures and the training data is not enough. Therefore, the model is prone to overfitting in the training process. Figure 8 shows the gesture categories in the dataset of Exercise A.

### 4.2. Method of Training

Adam [11] is used as the optimization method of network model training in this work. The length of data collected by each gesture in the two datasets was the same. After the same segmentation method, the data amount of each gesture was the same, and the samples are balanced. In the Myo dataset, each person has 2280 samples for each gesture, with a total of 19 participants, while in the Nina Pro dataset, each person has 1140 samples for each gesture, with a total of 10 participants.

At the same time, after the samples are segmented, we used the shuffle algorithm to shuffle the samples of each gesture, and then took 60% as a training sample set, 10% as the verification sample set, and the last 30% of each gesture as the test sample set. 

The amount of data fed into the network by each training batch was 128 samples, and a total of 50 rounds of iterative training were conducted. We initially set the learning rate at 0.01 and shrank the learning rate by 10 times at epoch 20 and 40. In order to prevent over-fitting of the network, L2 regularization is used in this paper. The parameter settings in the training process are shown in Table 2.

### 4.3. Myo Dataset Classification Results

The loss and accuracy curves during training and testing on the Myo Dataset using the EMGNet model are shown in Figure 9.

As shown in Figure 9, the EMGNet network successfully completed the classification task in the Myo Dataset, and the over-fitting phenomenon did not appear in the training and testing. We tested the accuracy of our model and compared it with the current three most advanced methods. Table 3 shows the accuracy of each method. According to the results shown in Table 3, the accuracy of our proposed model is better than the current advanced methods.

### 4.4. NinaPro DB5 Dataset Classification Results

Figure 10 shows that the loss and accuracy curves during training and testing on exercise A of the NinaPro DB5 Dataset. During training of the EMGNet model, the phenomenon of overfitting appears. Reducing the number of layers in the EMGNet does not solve this problem.

Table 4 shows the accuracy of three subsets of DB5 Dataset. Time domain characteristics (TD) [5], Enhanced TD [4], Nina Pro Features [6,22] and SampEn Pipeline [23] were selected as the classification features of the classical machine learning algorithm (LDA and SVM). The LCNN and CNN_LSTM we proposed previously did not perform feature extraction on the data and directly processed the sEMG signal. The proposed method uses continuous wavelet transform (CWT) to process the data as the input of EMGNet. From the experimental results, we can conclude that the classification accuracy of our proposed EMGNet model is higher than that of the classical machine learning algorithm.

With the increase of the categories of gestures, the classification algorithms decline at different degrees. The degree of decline of EMGNet is lower than that of the classical machine learning algorithms (see Figure 11 and Table 4).

The classification accuracy of both the classical machine learning method and EMGNet is not as high as that tested on the Myo Dataset (see Table 4). Two reasons are as follows:(1)The DB5 dataset has a relatively small amount of data per gesture and a relatively large number of gesture categories. For example, there are at most 7 gestures in the Myo dataset, while the smallest exercise A in DB5 dataset has 12 gestures.(2)There are a large number of similar gestures in the DB5 dataset. Figure 12 shows the sEMG signal spectrum of two processed samples in exercise A. The two samples shown in Figure 12c,d belong to different gesture categories, and it can be seen from Figure 12 that the channels with large fluctuations of the two gestures are almost the same. When the sample number of each gesture is insufficient, the model can easily misidentify it as the same gesture.

## 5. Conclusions

This paper presents a novel CNN architecture consisting of four convolutional layers and a max pooling layer with a compact structure and fewer parameters. The experimental results show that the proposed EMGNet not only reduces the complexity of the model but also improves the accuracy of the sEMG signal classification. It is highly competitive with the classical classifiers and deep learning frameworks currently used to classify sEMG signals, which also shows the potential of deep learning in classifying sEMG signals. (The elimination of transition time between successive gestures is explained in Appendix A).

## Figures and Tables

**Figure 1 sensors-20-00672-f001:**
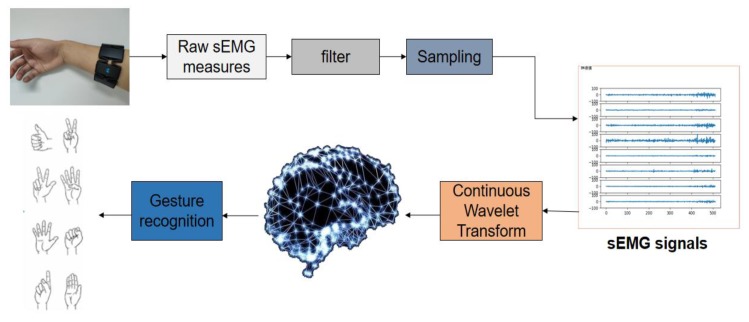
Surface Electromyography(sEMG) signals collection and classification process. Myo armband was used to collect the original signal, and then the collected signal was filtered and sampled to get sEMG signals. Continuous wavelet transform was selected to obtain the signal spectrum, and the neural network model was applied to classify the spectrum to achieve gesture recognition.

**Figure 2 sensors-20-00672-f002:**
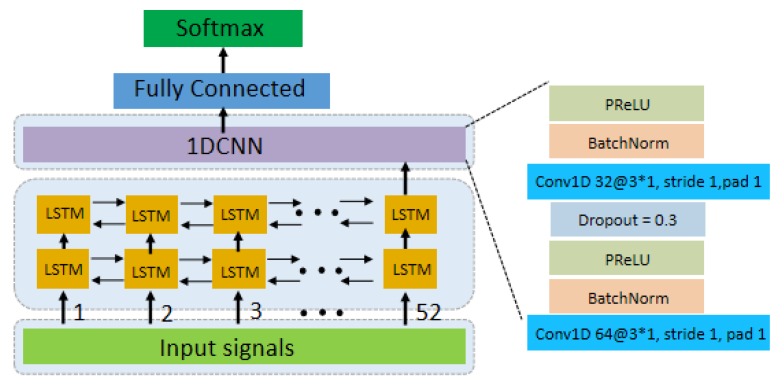
LCNN Architecture diagram, the LCNN consists of 2 LSTM layers, 2 one- dimensional convolution layers and 1 output layer. We use 2 LSTM layers, and each LSTM layer has 52 cells, and every cell has 64 hidden layers.

**Figure 3 sensors-20-00672-f003:**
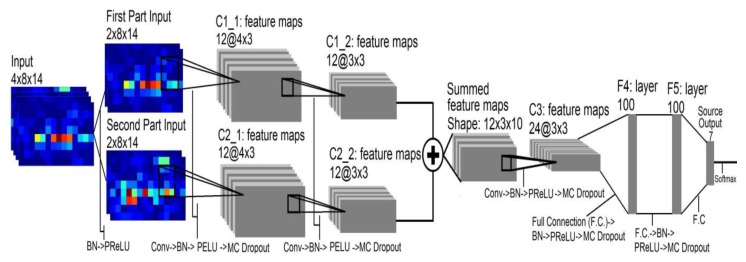
Schematic diagram of ConvNet architecture. In this figure, Conv refer to Convolution and F.C. to Fully Connected layers.

**Figure 4 sensors-20-00672-f004:**
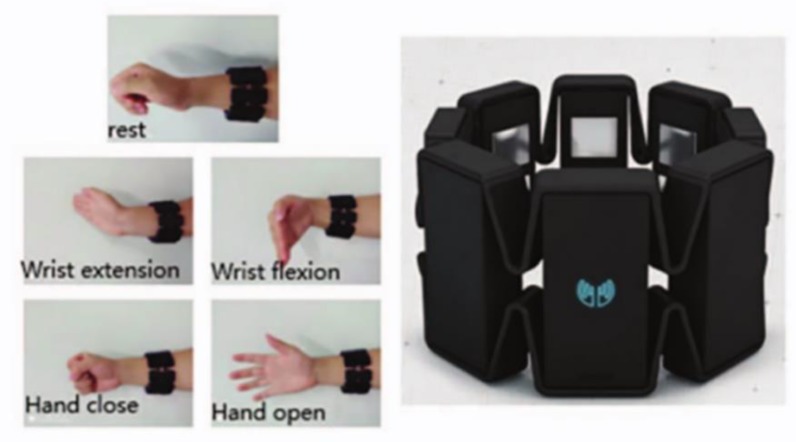
The 5 hand/wrist gestures and Myo armband. In this figure, the left is a schematic diagram of five gestures, and the right is the Myo armband.

**Figure 5 sensors-20-00672-f005:**
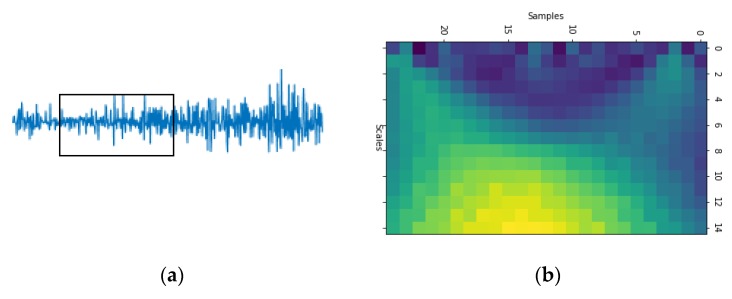
Part (**a**) shows the waveform of sEMG signals and Part (**b**) the spectrum of the sEMG signals shown in (**a**) after wavelet transformation.

**Figure 6 sensors-20-00672-f006:**
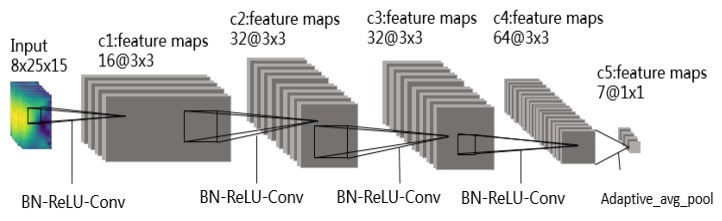
EMGNet Architecture contains four convolutional layers and a max pooling layer without using the full connection layer as the final output. In this figure, Conv refer to Convolution and avg_pool to a max pooling layer.

**Figure 7 sensors-20-00672-f007:**
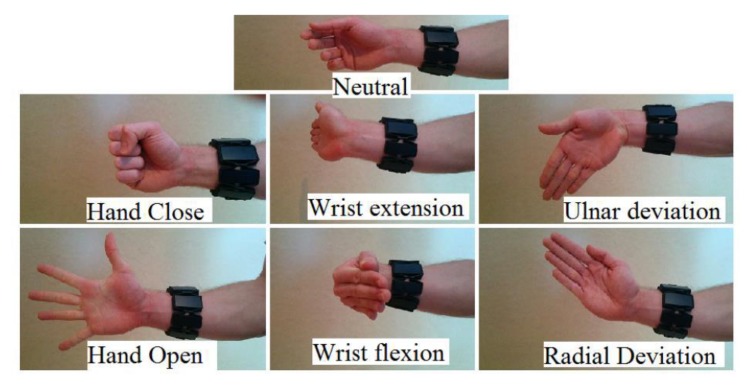
The 7 hand/wrist gestures in the Myo Dataset. In the Myo dataset, seven gestures are included: Netral, Hand Close, Wrist Extension, Ulnar Deviation, Hand Open, Wrist Flexion, and Radial Deviation.

**Figure 8 sensors-20-00672-f008:**
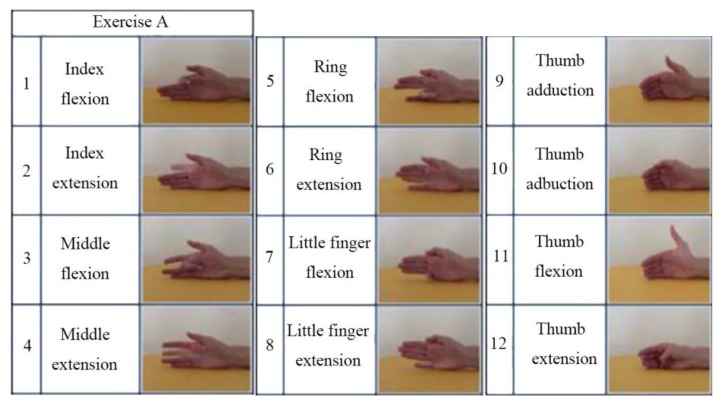
The gesture categories in the Exercise A dataset.

**Figure 9 sensors-20-00672-f009:**
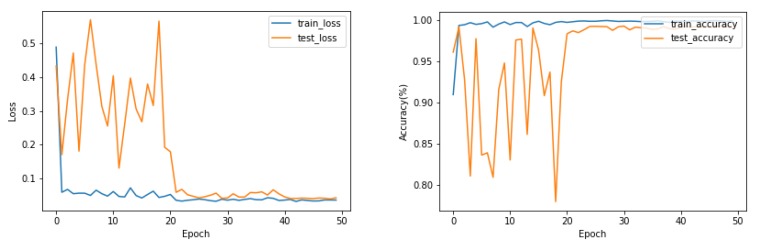
The loss and accuracy curves during training and testing on the Myo Dataset. when the training reaches convergence, there is no phenomenon where the accuracy of the training set is high and the accuracy of the test set is low, which is over-fitting.

**Figure 10 sensors-20-00672-f010:**
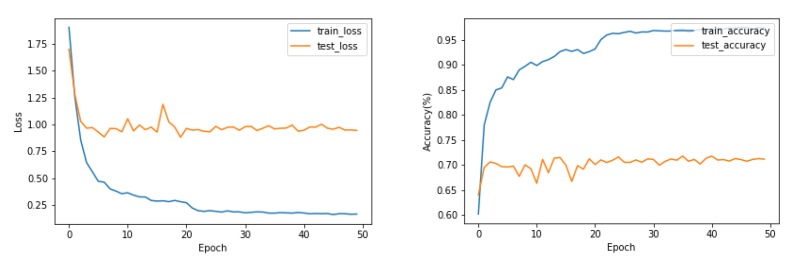
The loss and accuracy curves during training and testing on the exercise A of the NinaPro DB5 Dataset. During training and testing on the exercise A of the NinaPro DB5 Dataset, the phenomenon of overfitting appears.

**Figure 11 sensors-20-00672-f011:**
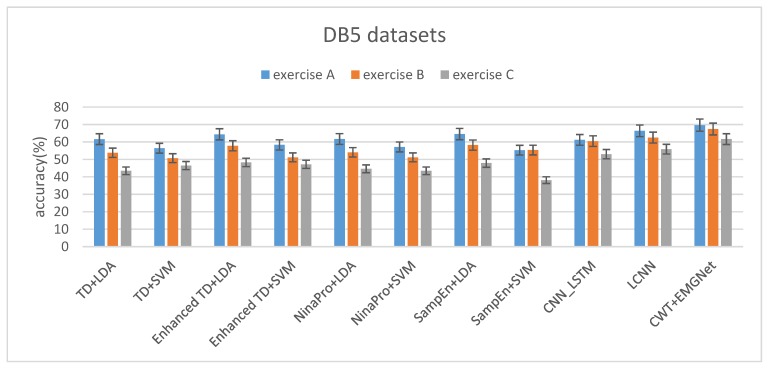
The average accuracy of three subsets on the DB5 Dataset.

**Figure 12 sensors-20-00672-f012:**
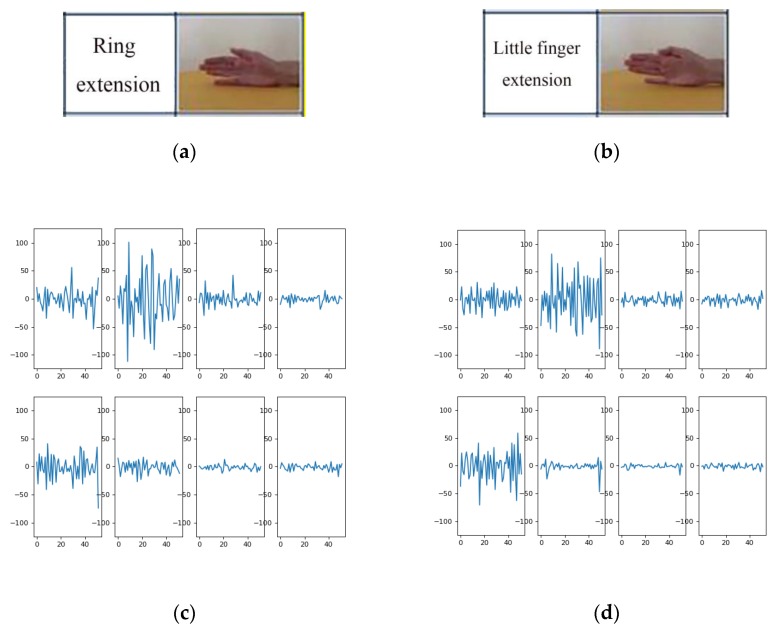
Parts (**a**) and (**b**) represent pictures of two different gestures in exercise A, Parts (**c**) and (**d**) show the spectrum diagrams of the sEMG signal generated by (**a**) and (**b**) respectively.

**Table 1 sensors-20-00672-t001:** Number of parameters used by various models.

Model	CNN_LSTM	LCNN	ConvNet	EMGNet
Number of parameters	56645	154353	67179	34311

**Table 2 sensors-20-00672-t002:** Training policy parameter setting (lr represents the learning rate).

Dataset	Initial lr	lr in 20 Epoch	lr in 40 Epoch	L2 Regularization
Myo Dataset	0.01	0.001	0.0001	1e-2
DB5	0.01	0.001	1e-3	1e-3

**Table 3 sensors-20-00672-t003:** Classification accuracy on the Myo Dataset. (We used the same samples to test these methods).

Model	TD+LDA	TD+SVM	Enhanced TD+LDA	Enhanced TD+SVM	NinaPro+LDA	NinaPro+SVM	SampEn+LDA	SampEn+SVM	LCNN	CNN-LSTM	CWT+TL	CWT+EMGNet
Accuracy (%)	97.54	96.61	98.16	96.40	97.54	94.73	98.02	95.95	97.88	96.77	98.31	98.81
STD (%)	2.60	5.50	2.30	4.70	2.60	6.70	2.10	5.70	2.20	2.80	2.16	1.51

^1^ The STD represents the standard deviation in accuracy for the 20 runs over the 17 participants.

**Table 4 sensors-20-00672-t004:** Classification accuracy on the DB5 Dataset. (We use the same samples to test these methods).

Model	TD+LDA	TD+SVM	Enhanced TD+LDA	Enhanced TD+SVM	NinaPro+LDA	NinaPro+SVM	SampEn+LDA	SampEn+SVM	CNN_LSTM	LCNN	CWT+EMGNet
EA(%)	61.61	56.40	64.36	58.32	61.66	57.12	64.53	55.28	61.21	66.38	69.62
EB(%)	53.80	50.70	57.80	51.17	54.05	51.18	58.20	55.34	60.46	62.51	67.42
EC(%)	43.48	46.49	48.26	47.18	44.61	43.46	47.89	38.12	52.99	55.87	61.63
$\frac{\mathrm{EA}-\mathrm{EC}}{\mathrm{EA}}$	29.4%	17.5%	25.0%	19.1%	29.5%	23.9%	25.8%	31.0%	13.4%	15.8%	11.5%

^1^ EA, EB and EC represent the accuracy of exercise A, exercise B and exercise C respectively.

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
