# Peer review of "Hand Gesture Recognition Using Compact CNN via Surface Electromyography Signals"

_sensors, 2020, doi:10.3390/s20030672_

Round 1

Reviewer 1 Report

In this paper, a new deep learning algorithm for gesture recognition is presented (EMGNet). Experiments on real data sets show how EMGNet obtains better classification performance than other methods.

It is not clear to me what EMGNet really classifies.

Suppose you have an online application and try to classify a consecutive set of gestures (A-> B -> C-> B -> .... etc.). Between each pair of gestures there is a transition from one position to the next position. The corresponding signal is very random and a transition from A to B is not the same as a transition from C to B. Even the same type of transition can be different.

A) Do you only classify gestures when the hand has reached a final gesture position? In this case, how do you detect transitions to avoid them? Or maybe, do you ignore them in your classifier by grouping all transitions as "no gesture"?

B) If you don't eliminate or ignore transitions, what does your algorithm do? Are you trying to integrate transitions into the "final positions" groups?

C) In relation to the learning process, do you have approximately the same number of gestures for each type? It seems that for Nina Pro, gesture sets are not well balanced, right? In this case, the deep learning algorithm will not have a good behavior. The same happens with other algorithms.

D) Suppose you have a balanced number of gestures for each type. If someone tries to use your algorithm, what would be approximately the minimum number of gestures of each type so that your algorithm is well trained?

Reviewer 2 Report

Reviewer’s Comment

In this manuscript, the authors claim that they have introduced a new deep neural network model for EMG hand gesture classification based that 1) requires a smaller parameter space 2) improves the accuracy of such classifications compared to the other models in the literature.

The manuscript lacks a well-defined structure. It is not clear what the goals of the authors are, how they are going about achieving them, and how these goals are related with the current problems and shortcomings that are faced while studying EMG for gesture recognition. The manuscript jumps between different topics without any coherent connection between them. In this reagards, it is particularly puzzling why the authors keep referring to image recognition literature? Did they make a use of specific step/strategy, etc. that is adapted from image recognition literatur? It is not clear how the cited studies are related to the present work and how the current manuscript help extend and/or improve these previous findings. The manuscript is not easy to read and it is quite confusing. The English of the manuscript requires extensive proofread and auditing.

I highlight some of the main issues that must be addressed before this manuscript can be considered for the review.

Title and Abstract: Please provide the full form for “sEMG” when it is first used. As a common practice, it is better to avoid the use of abbreviation in title.

Abstract: “By training the deep neural network model, the hidden features in sEMG signals can be extracted.” - It is better to start off the Abstract with this sentence to give a clear picture of what this manuscript is focusing on.

Abstract: “… the model often ...” - Which model? I presume that the authors meant the deep neural models in general. It is better to clearly refer to.

Abstract: “CNN” i.e., convolutional neural network suddenly appears here. It appears that the authors are using a specific type of DNN i.e., CNN. It is better if the authors clearly refer to this model (i.e., CNN) to avoid confusion and misinterpretation.

Introduction: “Distinguish sEMG signals is core part of related applications using sEMG signals as intermediate media.” - Distinguishing sEMG from what? Also, “Distinguish” should read “Distinguishing”.

Introduction: “Distinguish sEMG signals is core part of related applications using sEMG signals as intermediate media. At present, the literature on gesture recognition or artificial limb control by sEMG signals primarily focuses on feature extraction of sEMG signals, which aims to distinguish sEMG signals by feature recognition [1-3].” - This paragraph is not clear at all. What is to be distinguished? Why such a distinction is necessary? What are the features of interest? What these features are expected to signify/quantify?

Introduction: “… leading to the increasing of workload and dissatisfied result[14].” should read “… leading to an increase in workload and unsatisfactory results[14].” Moreover, it is not clear what do the authors mean by “increased workload”? In addition, the “dissatisfied results” claim by the authors appears to contradict their reference to “.. good results have been achieved ...” a sentence earlier.

Introduction: It is not clear why the authors refer to the application of deep learning in image recognition (I presume the aim of this manuscript is to use it on EMG data).

Introduction: “… which turns traditional design features into learning features.” I don’t understand what the authors try to express here. Are the authors referring to the use of variation of deep networks as autoencoders for automatic feature extraction?

Introduction, paragraph staring with “In recent years, deep learning ...” and ending by “can also reduce the demand for data size.” - This is very confusing. It is not clear at all what the authors try to accomplish. They cite a number of references that merely take a form of listing them. It is not clear what was the purpose of these cited studies and why they are included here. Most importantly, the authors appear to mix a number of issues (e.g., feature extraction and their appropriate selection, the effect of sample size and data scarcity, etc.) without explaining / discussing how these previous studies relate to them, address them or fail to address them adequately.

Introduction: “The main contribution of this paper is to present a new compact deep convolutional neural network (CNN) for gesture recognition, called as EMGNet.” - The lack of clarity of the precious paragraph about the previous findings and their shortcomings (please refer to my previous comment above) does not allow for clear understanding of why this is indeed a contribution and how and to what extent nt it may or may not help address the limitations that are currently faced while studying EMG.

Related Work: “Using convolutional neural network (CNN) to classify sEMG signals, literature [7,11,12] took the raw signals as input space.” - Do the authors mean that the earlier works made use of raw EMG data for classification of hand gesture? Or do they mean in any application domain of EMG (the authors earlier referred to its application – no cited literature - in limb control).

Section 3.1.: “In addition, WT, which is an ideal signal analysis tool, has abrupt amplitude and frequency, and its duration is limited, and the average amplitude of the entire duration is 0.” - It appears to me that the authors are mixing the advantages and disadvantages of WT in this sentence. It is not clear at all what this sentence is trying to convey: Whether it is a good idea to use WT or it is not!

Section 3.1.: “the mother wavelet function” - What is mother wavelet?

Section 3.2.: “which has been proved in a large number of literatures.” - This claim requires proper citations.

Section 3.2.: “Therefore, the size of all convolution kernels in EMGNet model is set to 3x3.” Why not 9X9 or 2X2? What is the significance of 3? How does it relate to previous sentence i.e., “which has been proved in a large number of literatures” Has it also been shown that 3 is an optimum choice? Did the authors find it to be a good choice? How did the authors go about finding this size?

Section 4.2.: “According to the results shown in Table 2, the accuracy of our proposed model is better than the current advanced methods.” These results correspond to 20 runs. Using the results of these runs, the authors can provide further insight on the observed improvement by applying a test of significant between the 20 results of these models. More importantly, the authors did not provide any information about the size of parameter space of these other models (except for CWT+TL in Section 3.2.). As a result, it is not possible to compare the observed differences in the improved accuracy and parameters trade-off. For example, if the authors can show that they significantly reduced the number of parameters (even though the improved performance was non-significant) or they significantly improved the performance (even though the parameters were not reduced significantly) then these results can be interpreted. Of course, the best case scenario would be to have both: significant reduction of parameter space along with a significantly improved performance. However, the latter is an ideal case. What is important is for the authors to show that their approach at least achieved one of these two.

Figure 9 (only the proposed methods and not the other techniques that are used to compare against): it is apparent the proposed method exhibited a high fluctuation in its accuracy on the test set among these 20 runs. Therefore, it is important to show the other methods’ breakdown as well.

Section 4.2.: With regards to the comparison of the proposed method’s performance, it is not clear whether the authors used the same test samples (i.e., in each of 20 runs) for all the models or not. If the test samples were not the same, it is not clear to me how the authors could compare the accuracy of these models.

Table 2 and Table 3: Why the authors reported different sets of results for Myo and DB5 datasets? Additionally, they provided accuracy-breakdowns for DB5 only (i.e., Figure 11).

Section 3.2.: “The model proposed in this paper only contains 34311 parameters.” This already looks like a large number of parameters. What is the typical number of parameters that are used in the literature for a similar task? Is the one used by the authors smaller? If so, is the difference significant? If positive, where these reduced number of parameters have mainly been applied and how this strategy appeared to make the proposed model introduced in this manuscript stand out compared to the previous models?

In general, there are many figures in this manuscript without sufficient explanation within the content of the manuscript.

Some of the grammatical issues are (but not limited to):

Abstract: “… model was validate ..” should read “… model was validated ...”

Abstract: “… the current state-of-the-art one.” should read “… the current state-of-the-art models that are used in this domain.” (I presume that the authors tested their model within the domain of hand gesture recognition and using Ninapro DB5 Dataset and the Myo Dataset).

Introduction: “… and robot technology,” should read “… and robotics technology,”

Introduction: “Rehman et al.[7] constructs ..” should read “Rehman et al.[7] construct …”

Introduction: “… put forward by [8,9]” should read “… put forward in [8,9]”

Related Work: “Since sEMG signals belongs” should read “Since sEMG signals correspond ...”

Related Work: “Figure 2 simply introduces the architecture...” should read “Figure 2 depicts the architecture ...”

Section 3.1.: “Slop Sign Changes” should read “Slope Sign Changes”

Section 3.1.: “SampEn” - I presume it is Sample Entropy. The authors must provide some short description of this and other features that are naming to help the reader have a better understanding of them and how their use may or may not be useful in the case of EMG.

Table 2. “standard variation” should read “standard deviation”

Round 2

Reviewer 1 Report

Ok about your answers.

However, I think you could comment in your document your criteria for selecting gestures and avoiding transitions. For example, the following sentences from your first answer: "We assume that each gesture can be maintained for a certain time", "... three sliding windows between four consecutive sliding windows ...." etc. explain how your algorithm makes a final decision.

Reviewer 2 Report

I thank the authors for addressing my comments. Although the manuscript has been certainly improved, in my opinion there are further changes and modifications needed.

First and foremost, the language of the manuscript requires a thorough auditing and proofread. Use of professional language editing services could be an option that authors can consider to further improve the quality of their manuscript.

With regard to the presentation and the results, my comments are as follow (please pay special attention to comment number 5).

1) Page 1: “...and some fruits have been achieved in ...”: Do you mean “some good results have been achieved”

2) The authors’ Response 8: “Since the sEMG signal was converted into an image in this paper, we use a neural network model to classify the converted image. Therefore, it is necessary to mention the development of deep learning in the field of image recognition in this article. ” - This point must be clearly mentioned in the manuscript to avoid confusion and also help the readers understand the reason why the authors are including image-ralated research in their manuscript.

3) Authors’ Response 9: Yes, we are referring to the use of variation of deep networks as autoencoders for automatic feature extraction. - Then please clearly mention that. For example: Wu et al.[14] proposed LCNN and CNN_LSTM autoencoders, which do not require ...

4) Auhtors’ Response 18: Since the parameters of the network model are randomly generated during the initial training process, the fluctuation of the test set loss curve and test accuracy curve shown in Figure 9 has a large randomness during the training process, so it cannot be compared with other methods. We can see from Figure 9 that when the training reaches convergence, there is no phenomenon where the accuracy of the training set is high and the accuracy of the test set is low, which is over-fitting. -

5)THIS IS VERY IMPORTANT - Given the response of the authors, I presume that the same test set has been used during the optimiztion steps of the model. In oder words, the results reported by the authors is not associated with an unseen data but the data that has been used for optimizing the trained model. Generally, this is not considered a valid way to test a model performance and the authors require to use more reliable verification step such as k-fold cross-validation.

With regard to the authors’ response that the performance of the other methods cannot be shown/used, I humbly disagree. These plots are related to the accuracy on the test set. As I mentioned above, first the authors need to correct their testing procedure by dividing their data into train, cross-validation, and test sets. Then, the authors can perform what has been shown in this plot (i.e., optimization of the trained model) using the train and cross-validation sets. Once the model is ready, the authors can use the trained model and apply it on the set aside test set and report the performance of their model on this unseen data. The authors can carry out these steps for k rounds (i.e., k-fold, for instance the same 20 runs that they have currently used) and report the results of their model versus other models in this fashion. The authors must also use these k runs for different models and apply a test of significance between all the models to show that their model in fact improve the accuracy (or number of parameters, etc.) compared to the other methods.

I also recommend that the authors present other performance metrics (e.g., precision, recall, and F1-score) and the confusion matrices of these models (i.e., their model along with the models that they used to compare their model’s performance with) to enable the readers have a more clear understanding of the model’s overall performance. Specifically, it is important for the authors to provide information whether the datasets that they used were balanced (i.e., equal number of gesture per gesture class) and also mention what the chance level accuracy is. Moreover, if the datasets include unbalanced classes then the authors must explain what step did they take to ensure the obtained accuracy is not due to only small portion of these classes that potentially have a larger number of samples. In general, it is important to ensure that the accuracy has been reported by considering the right proportion of all classes (e.g., 30% for testing must in fact include 30% of each class samples).
